# Oxidative Stress Is a Key Modulator in the Development of Nonalcoholic Fatty Liver Disease

**DOI:** 10.3390/antiox11010091

**Published:** 2021-12-30

**Authors:** Yuanqiang Ma, Gyurim Lee, Su-Young Heo, Yoon-Seok Roh

**Affiliations:** 1College of Pharmacy and Medical Research Center, Chungbuk National University, Cheongju 28160, Korea; yuanqiangma123@gmail.com (Y.M.); ssonhm117@gmail.com (G.L.); 2College of Veterinary Medicine, Jeonbuk National University, Jeonju 54896, Korea

**Keywords:** NAFLD, oxidative stress, ROS, mitochondria, ER stress, peroxisome, Kupffer cells, inflammation, hepatocytes, lipid metabolism, HSC, fibrosis, antioxidants

## Abstract

Nonalcoholic fatty liver disease (NAFLD) is the most common chronic liver disease worldwide, and scientific studies consistently report that NAFLD development can be accelerated by oxidative stress. Oxidative stress can induce the progression of NAFLD to NASH by stimulating Kupffer cells, hepatic stellate cells, and hepatocytes. Therefore, studies are underway to identify the role of antioxidants in the treatment of NAFLD. In this review, we have summarized the origins of reactive oxygen species (ROS) in cells, the relationship between ROS and NAFLD, and have discussed the use of antioxidants as therapeutic agents for NAFLD.

## 1. Introduction

### 1.1. Background of NAFLD

Nonalcoholic Fatty Liver Disease (NAFLD) is the most common chronic liver disease, and a large proportion of the population worldwide has it [1]. NAFLD is defined as an accumulation of fat (>5%) in the liver cells in the absence of excessive alcohol consumption or other causes of liver disease, including autoimmune disease, drug-induced conditions, or viral hepatitis [2]. Alcohol-associated Liver Disease (ALD) and NAFLD are the most common causes of chronic liver disease. Although they share a common spectrum of fatty liver/steatosis, steatohepatitis, fibrosis, cirrhosis and hepatocellular carcinoma, several differences exist [3]. Fatty degeneration of hepatocytes occurs more frequently in NAFLD than in ALD. In contrast, inflammatory cell infiltration and venous or intravenous fibrosis and venous sclerosis are more common in ALD than in NAFLD [4]. The spectrum of NAFLD ranges from simple steatosis (Non Alcoholic Fatty Liver or NAFL) to Nonalcoholic Steatohepatitis (NASH). NASH is characterized by inflammation, hepatocyte swelling, and varying degrees of fibrosis and has the potential to progress to cirrhosis and liver cancer (hepatocellular carcinoma (HCC)) [5]. Although the exact cause of NAFLD is not defined, recent studies have shown that oxidative stress (OS) due to insulin resistance and hepatic steatosis may be major causes of NAFLD and may play an important role in the progression to NASH [6]. The first stage of NAFLD/NASH is characterized by liver damage due to insulin resistance, increased fat load on hepatocytes [7]. Fat accumulation in the liver occurs as a result of an imbalance between the rate of influx and the rate of clearance of triglycerides (TGs). Most of the free fatty acids (FFAs) stored as TGs result from increased lipolysis in peripheral tissues. This increased lipolysis is a consequence of hyperinsulinemia and dietary fat that induce insulin resistance (IR), and increased lipogenesis. This can lead to inflammation, OS, lipid peroxidation, and mitochondrial dysfunction in the liver [8]. Therefore, fat accumulation in the liver causes metabolic disorders, leading to excessive mitochondrial ROS production and ER stress, which develops into inflammatory steatohepatitis (NASH). NASH is a pro-inflammatory state that leads to the activation of Kupffer cells (KCs) and astrocytes, which stimulate collagen deposition, leading to liver fibrosis [9,10].

### 1.2. Role of Oxidative Stress in NAFLD

Reactive oxygen species (ROS) are partially reduced oxygen metabolites and have a strong oxidative capacity [11]. ROS have many physiological activities in intracellular redox signaling and growth regulation [12]. Reactive nitrogen species (RNS) are various nitric oxide-derived metabolites, including the nitroxyl anion, nitrosonium cation, higher oxides of nitrogen, S-nitrosothiols, and dinitrosyl iron complexes [13]. ROS and RNS initiate, mediate and modulate intracellular OS through physiological (prohormonal action) or pathogenic (causing destructive vicious cycles) pathways [12]. The hydroxyl radical (HO¯) is the most reactive among free radicals and contributes significantly to the negative effects of OS. It damages biomolecules, induces lipid peroxidation, and induces breaks in DNA strands [12]. In physiological conditions, the equilibrium between the ROS, RNS, and antioxidants enables cellular crosstalk, control of intracellular functions, cell-to-cell interactions, proliferation, differentiation, migration, and contraction [14]. OS induced by ROS and inflammation are participants in the mechanisms that lead to hepatic cell death and tissue injury [15]. 

In this review, we summarize the pathophysiology of ROS-induced. We will also review the potential of antioxidants in the treatment of NAFLD.

## 2. Factors Contributing to Oxidative Stress in Liver

### 2.1. ER Stress and ROS

The endoplasmic reticulum (ER) is a fine network of tubules that performs many versatile functions in the cell [16]. The ER produces a vast majority of all proteins secreted into the extracellular space, including hormones and cytokines, as well as cell surface receptors and other proteins that interact with the environment [17]. In the ER, the catalytic processes of oxidoreductase Ero1 and NADPH oxidase (NOX) produce ROS. 

Reduced glutathione (GSH) is the most abundant thiol protein in eukaryotic cells and acts as a thiol–disulfide redox buffer. The balance of oxidized glutathione (GSSG) and GSH maintain intracellular redox homeostasis. In addition, the GSH/GSSG ratio represents the intracellular redox state index [18]. The oxidative activity of ER oxidoreductin 1 (Ero1) significantly contributes to cellular ROS [11]; its activity is similar to that of a number of ER oxidoreductases, including protein disulfide isomerase (PDI), endoplasmic reticulum protein (Erp)72, and Erp57 [19]. When GSH is oxidized, and the intramolecular level of GSSH is increased, Ero1-α is activated [20]. When Ero1-α is activated, it preferentially oxidizes the domain of protein disulfide isomerase (PDI). The oxidized a domain oxidizes the a’ domain in the molecule to a reduced state, and the reduced a domain of PDI is subsequently oxidized by Ero1-α to a fully oxidized and open form [21]. During oxidative protein folding in the ER, the thiol groups on the cysteines of substrate peptides are oxidized and form disulfide bonds, and hydrogen peroxide (H_2_O_2_) is generated as a byproduct (Figure 1). The oxidized a’ domain can be reduced by reduced ERp46 or GSH [19]. The a’ domain of the reduced PDI forms disulfide bonds [22], which contributes significantly to ROS production in cells [18]. A recent study revealed that the Ero1-α expression in the adipose tissue of homocysteinemia (HHcy) mice is upregulated and contribute to the accumulation of H_2_O_2_, ER peroxidation and ER stress, thereby exacerbating NAFLD [23]. 

NADPH oxidase (NOX) is the main enzyme responsible for ROS production. Growing evidence suggests that ROS production by NOX increases during ER stress [24]. The NOX family consists of Nox1–5 and Duox-1 and -2; these proteins promote the oxidation of NADPH to produce superoxide and have biological roles in various tissues [25]. Hepatocytes and hepatic stellate cells (HSCs) express NOX1, NOX2, and NOX4, whereas KCs predominantly express NOX2 [26]. Among the above, NOX4 is the most potent ROS-producing enzyme. NOX4 mainly produces H_2_O_2_. Unlike other NOX enzymes, NOX4 has an extended extracytoplasmic loop (E-loop), and because of this loop, the enzyme can be converted from an H_2_O_2_ -generating enzyme to O_2_¯—generating enzyme. A conserved histidine in the third extracytoplasmic loop of NOX4 is responsible for the H_2_O_2_-producing activity of the enzyme. Additionally, NOX4 interacts directly with p22 phox, and this interaction is a prerequisite for H_2_O_2_ production [27,28]. The conserved C-terminal dehydrogenase domain in NOX4 has constitutive electron transfer activity. In the full-length NOX4 protein, the N-terminal transmembrane portion of the enzyme generates ROS, and this has been shown to facilitate electron transport across the membrane on the opposite side of the membrane [29]. NOX4 has been studied as a ROS-generating factor in NAFLD. For example, a NOX4 inhibitor (GKT137831) was employed to confirm the role of NOX4 in NaAsO₂-induced HSC activation and ROS accumulation. The researchers reported that the expression of p22 phox, a subunit of NOX, was inhibited by GKT137831. Therefore, the inhibition of NOX4 was able to reduce the formation of NOX4-p22 phox dimers. In addition, the expression of collagen-1 and alpha-smooth muscle actin (α-SMA) was reduced by GKT137831. These results demonstrated that the activation of NOX4 can promote ROS generation and lead to HSC activation [30].

In conclusion, ER stress is induced through the misfolding of proteins in the ER, and ROS is generated by NOX during ER stress. The inhibition of ER stress signaling may reduce ROS generation, and this aspect can be targeted for developing new therapies for NAFLD.

### 2.2. Mitochondria and ROS

Mitochondria are important sources of intracellular ROS [31]. During cellular respiration, mitochondria transfer electrons to oxygen and generate ROS as a byproduct of oxidation [32]. Cellular stress such as hypoxia-reoxygenation and toxic substance processing can lead to excessive mitochondrial ROS (mtROS) production; the generated ROS are rapidly released into the cytoplasm, resulting in cellular damage [32]. mtROS are generated primarily during oxidative phosphorylation of electron transport chains (ETCs) present in the inner mitochondrial membrane [11]. Electrons donated from nicotine adenine dinucleotide (NADH) of ETC complex I (NADH dehydrogenase) and flavin adenine dinucleotide (FADH_2_) of complex II (succinate dehydrogenase) pass through ETC to complex IV (cytochrome c oxidase), which reduces O_2_ to water. On the other hand, protons (H⁺) actively migrate from the mitochondrial matrix to the intermembrane space, causing the mitochondrial matrix to increase in negative charge; this also increases the positive charge in the intermembrane space, creating a mitochondrial membrane potential (Δψₘ). This proton motive force causes complex V-ATP synthetase (ATP-ase) to generate ATP from adenosine diphosphate (ADP) and inorganic phosphates as the protons re-enter the mitochondrial matrix via complex V enzymes. During this process, leakage of electrons from complex I (NADH dehydrogenase) and complex III (ubiquinol-cytochrome C reductase) partially reduces oxygen to form peroxides (O_2_^•^¯). Subsequently, O_2_^•^¯ is rapidly transformed into H_2_O_2_ by two dismutases: superoxide dismutase 2 (SOD2) of the mitochondrial matrix and superoxide dismutase 1 (SOD1) in the mitochondrial intermembrane space. Collectively, O_2_^•^¯ and H_2_O_2_ generated in this process constitute mtROS (Figure 1) [33]. 

Another cause of mtROS production is an overload of mitochondrial calcium. Although Ca^2^⁺ is well-known as an important secondary messenger with a role in regulating many cellular physiological functions, Ca^2^⁺ overload is detrimental to mitochondrial functions and may contribute significantly to mtROS generation. Mitochondrial Ca^2^⁺ is involved in energy production (ATP), the opening of the mitochondrial permeability transition pore (m PTP), and in inducing and preventing apoptosis. 

In NAFLD, mtROS are oxidized to promote the cytoplasmic transport of mtDNA, and the oxidized mtDNA binds directly to NOD-, LRR- and pyrin domain-containing protein 3 (NLRP3) to stimulate interleukin (IL)-1β production [34]. ROS production is increased in HFD-fed NASH mice. In these mice, mtDNA is released into the plasma and contributes to the development of NASH by activating the toll-like receptor (TLR) 9. HFD-induced mtDNA release may increase the inflammatory response by activating the cyclic GMP–AMP-Synthase (cGAS)-cyclic GMP–AMP (cGAMP) stimulator of interferon genes (STING) pathway. This suggests that a high-fat diet contributes to the development of NAFLD by inducing OS and triggering the release of mtDNA, thereby accelerating liver damage [35].

Taken together, ROS in mitochondria are generated through ETC and Ca^2^⁺ overload. When mtROS is increased, a mitochondrial damage-associated molecular pattern (mtDAMP) such as mtDNA is released into and out of the cell due to mitochondrial dysfunction, which can accelerate NAFLD. Therefore, the maintenance of mitochondrial homeostasis through inhibition of mtROS can potentially be applied to treat NAFLD.

### 2.3. Peroxisome and ROS

Peroxisomes are multipurpose organelles involved in fatty acid α-oxidation, β-oxidation of very-long-chain fatty acids (VLCFA), purine catabolism, and the biosynthesis of glycerolipids and bile acids. Peroxisome-generated ROS accounts for about 35% of total intracellular ROS [36,37]. The main causes of ROS and RNS generation in the peroxisomes are D-amino acid metabolism and peroxisomal β-oxidation [36]. Peroxisomal β-oxidation is an essential regulator of hepatic lipid homeostasis. Acetyl-CoA, derived from peroxisomal fatty acid oxidation, plays an essential signaling role involved in the autophagic degradation of lipids. This suggests that the major source of cytoplasmic acetyl-CoA that maintains hepatic lipid homeostasis is peroxisomal β-oxidation [38]. The first step in the peroxisomal β-oxidation cycle is facilitated by FAD-containing acyl-CoA oxidase, which provides electrons directly to molecular oxygen to generate H_2_O_2_. This step begins in the classical peroxisome proliferator-activated receptor (PPAR)α regulatory and inducible β-oxidation helices dealing with straight-chain acyl-CoA, which in all species are catalyzed by a single enzyme, AOX. The disruption of the straight-chain acyl-CoA oxidase gene results in severe microvesicular hepatic steatosis in mice. This results in high levels of fatty acid chains in the serum, hepatomegaly, and steatohepatitis. This can accelerate the development of NAFLD [39]. The second and third steps of peroxisomal β-oxidation are catalyzed by two different dual-function proteins. The second step involves the hydration of enoyl-CoA to 3-hydroxyacyl-CoAs followed by and then dehydrogenation to yield 3-ketoacyl-CoA in the third step. Intracellular stimulation can also result in ROS-related effects on peroxisomes. Peroxisomes respond to a variety of external stimuli, such as fatty acids and pyruvates (such as peroxisomal proliferators); PPARα activation stimulates the proliferation of peroxisomes. Stimulated peroxisomes also show induction of the β-oxidation pathway; however, antioxidant enzymes are not simultaneously induced, which results in an increase in the intracellular H₂O₂ concentration (Figure 1) [40]. 

D-amino acid oxidase (DAO), which is highly expressed in the kidney and liver, is a flavin adenine dinucleotide (FAD)-dependent peroxisomal enzyme that catalyzes the oxidation of neutral and polar D-amino acids [41]. DAO catalyzes oxidative deamination and simultaneously reduces the cofactor FAD. When DAO is released, amino acids are hydrolyzed to 2-oxo acid (α-keto acid) and ammonia, and the reduced flavin is reoxidized to produce H_2_O_2_ [42]. The corresponding 2-oxo acids (α-keto acids), H_2_O_2_, and ammonia (the products of the D-amino acid oxidase reaction) are cytotoxic [42,43,44]. It has recently been shown that the D-amino acid oxidase/3-mercaptopyruvate sulfurtransferase (DAAO/3-MST) pathway is involved in the metabolism of hydrogen sulfide (H_2_S) [45].

In summary, ROS is generated through peroxisomal β-oxidation and DAO. Peroxisomal β-oxidation is an essential factor in the regulation of hepatic lipid homeostasis in NAFLD. Peroxisome-generated ROS may contribute to NAFLD progression.

## 3. The Mechanism of Oxidative Stress in Hepatic Steatosis

NAFLD progresses from simple steatosis to steatohepatitis, fibrosis, cirrhosis, and then to HCC due to insulin resistance, hepatic OS, and lipotoxicity mediated by a long-term western diet [46]. Simple hepatic steatosis occurs in the initial stages of NAFLD. There is a small amount of fat deposition, and, there, immune cells infiltration and hepatic cell damage are not present. Fat accumulation in more than 5% of hepatocytes leads to impairment in the metabolism system of the liver [47].

A “two-hit” strike theory was put forward by Day and James [48] to provide a theoretical basis for progression from NAFLD to NASH; this theory has been widely accepted. The “first hit” is manifested by simple hepatic steatosis. As the intake of free fatty acid increases triglyceride biosynthesis, it leads to fat accumulation in hepatocytes and a concomitant increase in insulin resistance. Moreover, the export efficiency of free fatty acids, TGs, and cholesterol is reduced severely. Thus, this forms the “first hit” of NAFLD. Oxidative stress is the initiator of the “second hit.” The first blow initiates metabolic derailment in the mitochondria, ER, and peroxisomes in hepatocytes. ROS can inhibit mitochondrial respiratory chain enzymes and inactivate glyceraldehyde-3-phosphate dehydrogenase and membrane sodium channels to induce hepatocellular injury [49]. ROS further aggravates lipid peroxidation, cytokines production and lipid accumulation and promotes inflammation and fibrosis through several protein kinases and nuclear transcription factor activation pathways.

### 3.1. The Mechanism of ROS-Mediated Oxidative Stress and Lipid Metabolism in Hepatocytes

ROS production is driven by the ETC in mitochondria as part of the energy production process. ROS play a role in redox biology and OS, and these two functions significantly contribute to physiological and pathological conditions. ROS at low concentrations [50] contributes to signaling transmission function [51], cell proliferation and differentiation [52], cell adhesion, and apoptosis. In contrast, high levels of ROS induce pathophysiological conditions. Loss of ROS homeostasis damages cellular lipids, proteins, and DNA since OS is induced by abnormal ROS levels [53]. Therefore, ROS play-acts as an essential role in the maintenance of physiological functions in the cell. Mitochondria are primarily responsible for the production of superoxide anions; thus, mitochondria are important metabolic and signaling hubs that function to maintain liver cell homeostasis, flexibility, and cell survival [54]. The metabolic burden of hepatocytes in a fatty liver is increased by a high intake of free fatty acids, cholesterol, and alcohol in the fatty liver. Dysfunctional electron transport and mitochondrial dynamics impair the lipid metabolism and lead to the downregulation of fatty acid consumption and upregulation of fat accumulation; this causes a further increase in mitochondrial ROS [55] and superoxide production [56]. The above has been confirmed in ALD [57] and NAFLD mouse models [58,59,60]. Thus, the disordered mitochondrial function involved in ROS production is a key factor affecting lipid metabolism.

Hepatocytes are parenchymal cells that constitute 80% of the liver mass [61], and are responsible for substance metabolism, detoxification, and protein production [62,63]. In ALD, NAFLD, and hepatitis C virus infections, ROS-mediated OS directly disrupts the function of mitochondria and ER and impacts fatty acid oxidation, lipid synthesis, and protein synthesis in the hepatocytes [54,64,65,66]. Therefore, OS plays an important role in the pathogenesis of liver disease. Hepatocytes contain 1000–2000 mitochondria to maintain normal metabolic functions [67]. Mitochondria are committed to β-oxidation, tricarboxylic acid cycle [68], ketogenesis, electron chain transfer activity, and ATP synthesis to maintain hepatocyte homeostasis [69]. In healthy hepatocytes, fatty acyl-CoA is transported into mitochondria (carnitine-based transport) and broken down to produce NADH and FADH2, which are further processed by the ETC to facilitate proton efflux, water formation, and ATP production. In NAFLD, fatty acid β-oxidation is induced because of increased fatty acid levels; therefore, ROS, which are byproducts of fatty acid metabolism, accumulate during the progression of β-oxidation and electron transport. Increased ROS in turn trigger insulin sensitivity, lipid metabolism disorders, inflammation, and apoptosis of hepatocytes. High ROS levels impact mitochondrial function, resulting in the blockage of the TCA cycle and fatty acid oxidation in the mitochondria. When the damaged mitochondria are unable to respond to the high levels of free fatty acids (the oxidative metabolism of fatty acids), the synthesis of fat is promoted in the cytoplasm. Insulin resistance promotes OS-driven steatosis. Lipid accumulation and OS collectively promote the development of NAFLD [70]. The maintenance of hepatocyte mitochondrial function is thus crucial for reducing steatosis in NAFLD. In the physiological liver microenvironment, cellular homeostasis is maintained by the clearance of damaged mitochondria by autophagy. Mitophagy is a type of autophagy pathway that helps maintain a constant number of mitochondria and is responsible for removing damaged mitochondria [55,71]. Mitophagy is important for clearing mitochondria damaged by ROS [72,73]. Many studies have focused on the PTEN-induced kinase (PINK)/parkin-dependent mitophagy signaling pathway [74,75]. Under normal conditions, stable PINK1 recruits parkin to mitochondria; parkin has been reported to promote the ubiquitination of mitofusin (Mfn1/2), voltage-dependent anion-selective channel (VDAC), and translocase of the outer membrane (TOM) proteins, and leads to enhanced mitophagy [76]. Low Δψₘ inhibits PINK1 degradation caused by presenilins-associated rhomboid-like (PARL) protein, which also induces mitophagy [76,77]. However, accompanied by excessive ROS production in mitochondria, oxidative stress and cell death are caused, which leads to a higher Δψₘ [78]. This indicates that the ROS-induced increase in Δψₘ is not conducive to the occurrence of mitophagy. A study found that the expression of parkin was decreased in NALFD; in contrast, inhibiting parkin degradation improved mitochondrial function by reducing ROS and malondialdehyde (MDA) levels, increasing antioxidant enzyme activity, and restoring mitophagy flux [79]. Insufficient mitophagy flux indirectly increases the accumulation of lipids driven by ROS in hepatocytes. In an HFD mouse model and an oleic acid (OA)/PA-stimulated cultured cell model, the loss of mitophagy was associated with fat accumulation, increased OS, and inflammation [80]. The fat accumulation, OS, and inflammation are alleviated by mitophagy and depend mainly on the mitochondrial damage mediated by ROS. In short, mitophagy plays a positive role as a protective mechanism against OS damage in hepatic steatosis (Figure 2).

### 3.2. Regulator of Oxidative Stress and Lipid Metabolism in Hepatocytes

OS occurs when the production of ROS and RNS is very high. Redox reactions of H_2_O_2_, O^2−^ or OH^−^ ions with DNA, proteins, and lipids within cells causes oxidative injury [81]. AMP-activated protein kinase (AMPK) is involved in the regulation of a variety of cellular metabolic pathways. AMPK inhibits the activity of lipid metabolism-related enzymes such as acetyl-CoA carboxylase (ACC) and transcription factors such as PPARα, PPARγ, and sterol regulatory element-binding proteins (SREBPs); this alleviates lipid accumulation and promotes fatty acid oxidation [82]. AMPK signaling activity is sensitive to ROS, as both endogenous and exogenous ROS can activate AMPK [83,84]. However, in response to H_2_O_2_, ataxia–telangiectasia mutation (ATM) activates the tuberous sclerosis complex 2 (TSC2) via AMPK, which results in the inhibition of mammalian targets of rapamycin (mTOR) and the induction of autophagy. This indicates that ROS activates AMPK but exacerbates cell damage. For instance, Fudan–Yueyang–Ganoderma lucidum (FYGL), extracted from *Ganoderma lucidum*, increased AMPK and ACC activity and ameliorated an imbalance in oxidation and anti-oxidation to promote a steady state. Thus, AMPK is essential for the maintenance of lipid metabolism in hepatocytes.

H_2_O_2_ is a strong oxidizing agent and stimulates lipid accumulation through SREBP1c in vitro [85]. In NAFLD, the upregulation of SREBP1c is associated directly with H_2_O_2_-triggered OS. In turn, the overexpression of SREBP1c also enhances ROS levels in hepatocytes, increases lipid synthesis, and activates the NF-kB inflammation pathway [86]. The overexpression of SREBP-1 in prostate cancer cells promoted ROS generation, fatty acid synthase expression, and accumulation [87]. This evidence indicates that ROS promotes hepatic steatosis and is an essential mediator in the development of this condition. The inhibition of ROS is considered an important strategy for the alleviation of lipid accumulation in hepatocytes.

Redox homeostasis mechanisms include oxidant and antioxidant systems, and these systems regulate ROS clearance in many cell types. Antioxidant enzymes such as superoxide dismutase (SOD), catalase (CAT), and glutathione peroxidases (GPxs) in the cell are primarily responsible for lowering ROS levels [88]. ROS can be neutralized by SODs, which catalyze their conversion to the less active H_2_O_2_ molecule, which is further reduced to water by the CAT and GPx enzymes [53,89]. However, enzymes involved in lipogenesis and lipid droplet accumulation were elevated in the cultured SOD1-deficient hepatocytes [90]. A similar mechanism involves carnitine palmitoyltransferase 1 (CPT1); this enzyme is involved in fatty acid metabolism and is positively regulated by SOD1 to prevent lipid accumulation in nasopharyngeal carcinoma [91]. Unlike SOD1, mitochondrial SOD2 is a key component that regulates ROS metabolism in the mitochondrial matrix. The elimination of excessive ROS depends on SOD, and this process is essential to alleviate lipid accumulation and metabolic disorders in damaged hepatocytes.

In NAFLD, abnormal ROS metabolism leads to an imbalance between OS and anti-OS mechanisms in hepatocytes. Nuclear factor erythroid 2-related factor 2 (NRF2) is an antioxidant transcription factor that negatively regulates ROS-mediated OS [92]. Other evidence indicates that NRF2 is also involved in lipid metabolism. Increased NRF2 expression in the nucleus induces antioxidant gene expression and inhibits ROS production induced by FFA; this, in turn, blocks lipogenic transcription factors, such as CCAAT-enhancer-binding proteins (C/EBP)α and PPARγ [93]. However, NRF2(L)-KO mice with liver-specific NRF2 knockout on an HFD showed less liver enlargement, inflammation, and hepatic steatosis compared to the wild-type mice; NRF2 (MHFD macrophage-specific NRF2-knockout) mice on an HFD were comparable to the wild-type mice in this regard and showed no significant differences in these parameters [94]. This is most likely due to the dual role of NRF2. Chambel et al. indicated that activated NRF2 inhibited lipid accumulation in white adipose tissue, limited adipogenesis, induced insulin resistance and glucose intolerance, and increased hepatic steatosis in ob/ob mice [92]. The differences in the ages of the experimental animals may explain the contradictory findings in the studies. In summary, the antioxidant mechanism involving NRF2is important for the elimination of cellular ROS (Figure 2).

### 3.3. Oxidative Stress-Mediated Hepatocytes Apoptosis

High ROS levels lead to the loss of function of various intracellular organelles such as mitochondria and ER, which can result in the activation of apoptosis and cell death. OS-mediated hepatocyte apoptosis and dysfunction increase the infiltration of KCs and the activation of HSCs during liver damage [95]. ROS cause hepatocyte dysfunction by damaging intracellular macromolecules, including lipids, proteins, and DNA. Additionally, a study based on an in vitro menadione-generated superoxide model reported that ROS could directly activate c-Jun N-terminal kinase (JNK)/c-Jun cell death signaling pathways to induce hepatocyte apoptosis [96]. Importantly, hepatocytes are highly susceptible to mitochondrial injury and cell apoptosis in the presence of diminished antioxidant ability. The overproduction of ROS destroys mitochondrial proteins, phospholipids, and mitochondrial DNA (mtDNA) [97,98]. The depletion of mtDNA in hepatocytes leads to the reduced expression of mitochondrial DNA-encoded polypeptides and thus results in mitochondrial dysfunction [99]. Mitochondrial dysfunction promotes the progression of hepatocyte apoptosis. Apoptosis and necrosis of hepatocytes are obvious features of NASH. Hepatocyte apoptosis was reported to correlate positively with hepatic fibrosis [100]. Research indicated that OS and apoptosis were augmented, and the proliferation of hepatocytes was reduced in high cholesterol-fed mice [101]. It is possible that hepatocytes with loss of organelle function are unable to process lipid droplets and cholesterol adequately, leading to lipid accumulation.

## 4. The Mechanism of Oxidative Stress of Kupffer Cells in NASH

In simple steatosis, lipotoxicity and fat accumulation cause severe inflammation in the liver, which leads to NASH. Liver macrophages are mainly two types: monocyte-derived macrophages, which are transported into the liver through blood circulation, and liver-specific resident macrophages, known as KCs (Kupffer cells), which are derived from bone marrow progenitor cells (constitute almost 80–90% of all liver macrophages) [60,102]. KCs play an important role in the liver immune response and are also responsible for ROS generation through the activation of NOX2 and TLR signaling [103]. Of note, ROS are key signaling molecules in the progression of the inflammatory response. In NAFLD, the accumulation of fat leads to an increase in the endotoxin/lipopolysaccharide (LPS) levels in the liver, which induces the M1 polarization of KCs. The M1 polarized macrophages produce ROS and inflammatory cytokines. The pro-inflammatory cytokines secreted from macrophages induce infiltration of neutrophils and the activation of HSCs, which aggravates liver damage and then leads to the progression of hepatic fibrosis.

### 4.1. mtDNA Mediated Kupffer Cells Activation in NASH

OS-mediated hepatocyte apoptosis and dysfunction increase the infiltration of KCs and the activation of HSCs during liver damage [95]. This exacerbates the progression from NAFLD to NASH and HCC. Steatosis promoted by OS is the main cause of liver cell apoptosis. ROS prevent the replication and transcription of mtDNA, which leads to mitochondrial dysfunction, and further causes increased ROS production and further mtDNA injury [104]. mtDNA is released from the liver cells and is recognized by the innate immune cells (including resident KCs and monocyte-derived macrophages), leading to an inflammatory response; this further promotes the progression of NASH.

In general, NAFLD ranges from simple steatosis to NASH, and the pathogenesis of NASH is still unclear. Recent research has revealed high serum levels of mtDNA and mitochondria in mouse models and NASH patients. KCs activated by mtDNA secrete pro-inflammatory factors (tumor necrosis factor (TNF)-α and IL-6) through TLR9 and stimulate IFN genes (STING) [105,106,107]. STING is a universal receptor for cyclic dinucleotides, including the bacterial second messengers 3′-5′-cyclic-di-adenosine-monophosphate (CDA), 3′-5′-cyclic-di-guanosine-monophosphate (CDG), and the newly discovered metazoan second messenger ring GMP–AMP [108]. The STING pathway is activated by mtDNA and operates in the liver as well as in other tissues, such as in the lungs (in sepsis-induced acute lung injury) and kidneys (in acute kidney injury) [109,110]. In the NASH mouse model, the lack of STING alleviates hepatic steatosis, fibrosis, and nuclear factor (NF)-κB-dependent inflammation even in the presence of mtDNA-induced stimulation of KCs. The mtDNA derived from hepatocytes is a vital factor that drives the innate immune responses of KCs and the activation of HSCs to accelerate NASH and hepatic fibrosis (Figure 3). As mentioned above, the release of mtDNA constitutes a DAMP and is a response to mitochondrial dysfunction caused by ROS-mediated OS in hepatocytes. A previous study found that KCs also reduced the Δψₘ under the stimulation of saturated fatty acids (palmitic acid) and increased the release of mtDNA from the mitochondria to the cytoplasm, leading to an increase in the expression of NLRP3 inflammasome and the secretion of IL-1β [111]. The burst of OS induces the release of mtDNA from hepatocytes or KCs itself; this event is important for KC activation through the STING pathway.

### 4.2. Oxidative Stress Mediated Inflammation in Kupffer Cells

In ALD and NAFLD, intestinal endotoxin/LPS is the main stimulator of KC activation, which results in the production of pro-inflammatory factors, chemokines and abundant ROS via TLR4 receptor signaling [65,112]. Although hepatocytes and KCs are the main sources of ROS production, hepatocytes have a higher antioxidant capacity than KCs [113]. It is possible that KCs may have a greater impact on the ROS-mediated mechanisms compared to hepatocytes. The reason may be the high expression of uncoupling protein 2 (UCP2) in hepatocytes and the reduction in KCs in HFD-fed mice [114,115]. UCP2, a mitochondrial inner membrane protein, reduces the Δψₘ and prevents OS-mediated damage by increasing proton flux and ATP synthesis [116]. Research confirms that when Δψm is dissipated by chemical uncoupling agents or overexpression of mitochondrial uncoupling proteins, ROS production is reduced [33,117]. LPS treatment of macrophages with a targeted disruption of UCP2 led to ROS over-production and pro-inflammatory cytokine secretion through NF-kB pathway activation [118]. Therefore, UCP2 hinders the activation of KCs, and this depends on the negative regulation of Δψₘ and ROS levels. Apart from ROS generated by dysfunctional mitochondria, peroxisomes and the microsomal cytochrome P450 system also generate ROX, which affect redox metabolism in KCs.

When KCs and other macrophages in the liver respond to LPS or other stimuli, they rapidly release ROS through the specialized phagocyte oxidase gp91phox or NOX2 systems [119]. The activated NOX2 produces superoxide, which sends signals to thioredoxin, protein kinase C, extracellular signal-regulated kinase (ERK) family members, and NF-κB to respond to the stimulus [120]. Thus, ROS increase the production of pro-inflammatory factors such as TNF-α, IL-6, and IL-1β. NOX2 activation involves a series of protein–protein interactions; phosphorylation of p47phox allows it to interact with p22phox that is constitutively bound to NOX2. p47phox then recruits additional proteins such as p67phox, p40phox, and Rac GTPases to form a complex with NOX2, which leads to NOX2 activation [121]. This protein complex with activated NOX2 generates superoxide ions by transferring an electron from NAPDH to oxygen. The gene ablation of NOX2 leads to a decrease in ROS production, and attenuates M1 activation and promotes M2 polarization of macrophages in the liver.

FFAs and cholesterol contribute to the formation of fatty “foamy” KCs, and can further lead to NASH [122]. FFAs directly promote the M1 polarization of macrophages that results in a pro-inflammatory phenotype [123,124,125]. Excessive FFAs not only aggravate lipid accumulation in hepatocytes but also damage β-oxidation and mitochondrial function in KCs [125]. Studies showed that KCs sense FFAs and promote lipid accumulation in hepatocytes via TNF-α secretion [126]. It is common knowledge that foam cell formation is the main event in the pathogenesis of atherosclerosis [127]. Interestingly, ROS also promote the formation of foam cells [128], and foam cells were also found in the liver of a NAFLD mouse model [129]. Foam cells in the liver may arise from liver-resident KCs, infiltrating macrophages, or a combination of both the cell types [130]. However, whether ROS participates in the formation and regulation of foam cells and during NAFLD remains unclear.

In the antioxidant system, antioxidant proteins and phase II detoxification enzymes (heme oxygenase-1 (HO-1) and NADPH-quinone oxidoreductase-1 (NQO1)) are key protective agents against oxidative damage [131]. Studies have found that the upregulation of HO-1 induced by ethanol involves the activation of JNK-1, hypoxia-inducible factor (HIF)-1α and NRF2, but the expression of NQO1 is only regulated by NRF2 [129]. By upregulating antioxidant genes, NRF2can also protect the liver from inflammation in a ROS-dependent manner [132]. Although NRF2 is elevated under external stimulation, it is downregulated in NASH patients [133]. A recent study determined that inflammation and metabolic deterioration were aggravated in a myeloid-specific NRF2-deficient NASH mouse model [134]. In another study, LPS accelerated the inflammatory response in P62 and NRF2-double knockout KCs [135]. The antioxidant system is indispensable for the inflammatory responses of KCs (Figure 3). This system balances the redox state and alleviates the inflammatory responses mediated by ROS.

## 5. The Mechanism of Oxidative Stress in Hepatic Fibrosis

The accumulation of fat within the liver triggers chronic inflammation mediated by activated KCs and other infiltrating immune cells; this leads to hepatic fibrosis. About 9% to 25% of NASH patients will develop cirrhosis in 10 to 20 years [136]. Hepatic fibrosis is characterized by the activation of HSCs; this leads to the proliferation and migration of HSCs and results in a fibrotic phenotype (myofibroblasts) [137]. The activated HSCs further promote the formation of excess collagen and the accumulation of extracellular matrix (ECM). TLR activity [138], autophagy [139], ER stress [140], OS [141], and extracellular signals from hepatocytes and KCs and other molecular signaling pathways have been reported to be involved in the activation of HSCs [142,143]. In this section, we focus primarily on the mechanism of HSC activation mediated by OS.

### 5.1. ROS-Mediated Fibrogenesis in Stellate Cells

OS results from the alteration of the oxidation and antioxidant balance and involves changes in ROS and RNS levels. ROS not only increase hepatocyte metabolism disorders and KC activation but also stimulates the production of collagen I, which is a key intracellular signaling mediator in the TGF-β-driven fibrosis [144]. TGF-β is an important cytokine that induces the activation of HSCs. Under normal conditions, TGF-β participates in wound healing and angiogenesis. However, excess TGF-β is released from damaged hepatocytes and pro-inflammatory KCs [145,146]. This released TGF-β is recognized by the TGF-βRI receptor on the HSCs and increases fibrogenesis via the SMAD2/3 pathway [141]. In this process, phosphorylated SMAD 2/3 also activates the expression of NOX4 [147,148]. Interestingly, NOX enzyme complexes and the mitochondria are the main producers of endogenous ROS [149]. Moreover, studies have found that the expression of NOX4 is upregulated in many fibrosis models [150]. NOX4 knockdown reduces the activation of HSCs without affecting the expression of the TGF-β1 receptor and the phosphorylation of SMAD2, which indicates that NOX4-mediated ROS occur downstream of the TGF-β1 and SMAD complex [151]. The downregulation of HSC activation depends largely on the lack of NOX4 (which indicates a lack of excessive ROS). In addition to NOX4, NOX1 and NOX2 are also expressed in HSCs. The depletion of NOX1 or NOX2 can reduce inflammation and fibrosis caused by carbon tetrachloride (CCl_4_) and bile duct ligation (BDL) [152,153]. In contrast, NOX4 does not need to recruit cytoplasmic structural subunits to form active enzymes to generate ROS [64]. This implies that NOX4 produces ROS more rapidly than NOX1 and NOX2. Surprisingly, H_2_O_2_ and other ROS can directly induce collagen α1 activity [145,154]. Therefore, OS due to ROS plays a key role in TGF-β-mediated fibrogenesis in HSCs (Figure 4).

A study has found that the components of the NLRP3 inflammasome are expressed in HSCs and contribute to the activation of HSCs [155]. NOX4-mediated ROS production activates the NLRP3 inflammasome and hinders the antioxidant system, leading to HSC activation and fibrosis [156]. TGF-β-induced ROS produced by mitochondria can stimulate NLRP3 activation to increase the levels of α-SMA, connective tissue growth factor (CTGF), and tissue inhibitor matrix metalloproteinase 1 (TIMP1), whereas knockout of p66Shc, an oxidoreductase, eliminated this effect [157]. A recent study indicated that the knockout of NLRP3 downregulated ROS production and the mRNA and protein levels of fibrosis markers in HSCs [158]. These data indicate that ROS increases the activation of fibrosis through the activated NLRP3 inflammasome. However, these data do not rule out the role of secreted mature IL-1β and IL-18 in fibrosis.

### 5.2. The Role of Antioxidants System in Stellate Cells

Hepatic fibrosis is characterized by chronic liver damage caused by excessive deposition of ECM-collagen initiated by HSCs. Platelet-derived growth factors (PDGF) and TGF-β1 are key mediators that trigger the activation of HSCs. Moreover, OS is involved in the cytokine-mediated activation of HSCs and in the process of liver fibrosis. Certainly, the antioxidant system acts to remove intracellular ROS and maintains the normal physiological activity of HSCs. In many cell types, NRF2, which is the most important antioxidant transcription factor in the cell, transcriptionally modulates more than 500 genes, most of which are involved in cell protection mechanisms [159]. These antioxidant genes include SOD, CAT, GPx, GSH, HO-1, and NQO-1 [160]. Oxidative or electrophilic stress causes the dissociation of Keap1 from NRF2, followed by the nuclear translocation of NRF2; NRF2 subsequently modulates gene expression in the nucleus [161]. These genes are transcribed to block the activation of HSCs mediated by ROS such as H_2_O_2_, O^2−^ and OH^-^; this alleviates the accumulation of ECM [160]. During the development of NASH, the NRF2-mediated antioxidant system is unable to adequately limit the ROS overload stimulated by PDGF and TGF-β1 (secreted by KCs and HSCs). Antioxidant targeted- NRF2 increases the expression of heme oxygenase 1 (HMOX1), NQO1, and glutathione S-transferase mu3 (GSTM3) genes, and downregulates NOX1, NOX4, and α-SMA in activated HSC models [162,163]. In NRF2-deficient HSCs, the expression of α-SMA and ECM increases, which involves the upregulation of the TGF-β1/ SMAD pathway [159]. Thus, NRF2 is a potential therapeutic target to prevent the activation of HSCs by ROS (Figure 4).

## 6. Antioxidants as Treatment for NAFLD

### 6.1. Flavanols (flavan-3-ols)

Flavonoids are divided into several subtypes according to the substitution pattern on the C ring, which depends on the degree of oxidation of the carbon to which the ring is attached [164]. Flavonoids are mainly the following types: flavones, flavonols, flavanones, flavanonols, isoflavones, and flavan-3-ols [165]. These compounds are known to have hepatoprotective effects based on their anti-inflammatory, antioxidant, blood-sugar-lowering, and hepatic lesion suppression activities. They also function as scavengers of numerous active substances such as peroxide and hydrogen peroxide; CYP2E1 is one of the molecules responsible for the production of these active substances [166]. Flavanols such as catechins have been shown to exert several beneficial effects in NAFLD. Epigallocatechin-3-gallate (EGCG) is a catechin found in green tea [167]. The effect of EGCG intake on GSH levels, lipid peroxidation, and the expression of CYP2E1 was studied in an HFD mouse model. The results demonstrated that EGCG improved fat accumulation and inflammation in the HFD mice [168]. The effect of EGCG in inhibiting HSC activation has been demonstrated in several in vitro studies. A recent study showed that intraperitoneally administered EGCG could reduce liver fibrosis in a rodent model of NASH by inhibiting NF-kB, Akt, and TGF/SMAD signaling as well as OS [169]. EGCG significantly decreased the levels of ROS and MDA in rat serum and liver tissue and increased SOD (Mn SOD, Cu/Zn SOD) enzyme activity and the mRNA levels of GSH-Px. In addition, EGCG decreased the levels of inflammatory factors and the mRNA expression of TNF-α and IL-6 in serum and liver tissue [170].

### 6.2. Curcumin

Curcumin is a natural polyphenol with antioxidant and anti-inflammatory properties [171]. The antioxidant properties of curcumin have been extensively studied. This compound has been reported to have positive effects on NAFLD and liver metabolism [172]. In a recent study, curcumin effectively reversed the expression of CYP3A and CYP7A in the fatty liver, restoring metabolic capacity [173]. In addition, curcumin exerts a protective effect against inflammatory stimuli by downregulating the expression of inflammation-related transcription factors and reducing the activity of related signaling pathways against linoleic acid and leptin-induced stimuli. Curcumin has a prophylactic effect by targeting circulating monocytes, hepatic macrophages, and peripheral and hepatic CD4+ cells. Curcumin administration inhibited NAFLD development in a mice model of HFD-induced obesity and glucose intolerance and ameliorated histological changes, including fibrosis and intrahepatic accumulation of CD4+ cells [174]. Curcumin exerts beneficial effects on fatty liver disease by reducing hepatic lipid accumulation through the modulation of the AMPK/ACC pathway. *O*-GlcNAcylation was increased in mice fed the methionine and choline diet (MCD); the upstream target inositol-requiring transmembrane kinase/endoribonuclease 1α (IRE1α) was induced by ER stress, and this factor further upregulated N-acetylglucosaminyltransferase (OGT) and glutamine: fructose-6-phosphate amidotransferase (GFAT).

These results suggest that the induction of OGT and GFAT by the transcriptional regulation of X-box binding protein 1 (XBP1) (which is as an upstream activator of hexosamine biosynthetic pathway (HBP)) activates a positive regulatory loop in NASH pathogenesis. Curcumin effectively inhibited *O*-GlcNAc protein modification and further downregulated ER stress by inhibiting XBP-related IRE1α and GFAT expression [175].

### 6.3. Vitamin E

Vitamin E is a lipid-soluble chain-breaking antioxidant (breaks the chain of lipid oxidation) that prevents the production of free radicals. Vitamin E can only be synthesized by plants and exists in the form of four fat-soluble tocopherols and eight forms of tocotrienols that exhibit antioxidant activity [176]. This vitamin has unique antioxidant activity, and the chromanol ring in its structure provides hydrogen ions to free radicals, which results in lipid peroxyl radical scavenging activity. It also increases the activity of other antioxidant enzymes such as catalase and glutathione peroxidase [177]. α-Tocopherol is the most abundant form of vitamin E in nature and the most common and dominant form in human tissues and plasma [178]. In obese (ob/ob) mice, α- or γ-tocopherol showed a protective effect against LPS-induced liver damage and decreased the expression of liver MDA and TNF-α [179]. In addition, vitamin E improved liver necrosis when CCl_4_ was administered to rats. Vitamin E downregulated the expression of TGF-β1, which is a cytokine implicated in the pathogenesis of liver fibrosis [180].

### 6.4. Metformin

Metformin (N,N-dimethylbiguanide) belongs to the biguanide class of antidiabetic drugs. Several studies have reported that metformin treatment slows the progression of NAFLD [181]. Metformin has been reported to protect HepG2 cells and primary murine hepatocytes from palmitate- and stearate-induced apoptosis. Metformin partially inhibited mitochondrial respiration, restored Δψₘ, decreased ROS production, and induced SOD2 expression. Metformin partially inhibits mitochondrial complex I, thus preventing mitochondrial dysfunction; it also protects the cell from palmitate-induced cell death [182]. In addition, four weeks of metformin treatment (50 or 100 mg/kg) in obese mice also reduced hepatocyte apolipoprotein A5 (ApoA5) expression; this protein plays a key role in the regulation of TG metabolism and lipid droplet formation [6]. In another study, metformin (200 mg/kg) significantly reduced plasma TG levels, decreased hepatic very-low-density lipoprotein (VLDL)-TG production, and lowered hepatic lipid composition in the APOE*3-Leiden CETP mouse model [183]. Interestingly, metformin (250 mg/kg) enhanced AMPK activation. Metformin-catalyzed increased AMPK phosphorylation suppressed SREBP-1c expression, thereby regulating lipid and glucose metabolism [184]. AMPK also activated NRF2 by directly phosphorylating it or by inhibiting glycogen synthase kinase (GSK)3β, an inhibitory regulator of NRF2 [185].

### 6.5. Coenzyme Q_10_

Coenzyme Q_10_ (CoQ_10_) is a fat-soluble active quinone that contains a benzoquinone ring with an isoprenoid side chain. CoQ_10_ is distributed in organelles such as the Golgi apparatus, ER, and lysosomes; 40–50% of the total cellular content of coenzyme Q_10_ is localized in the inner mitochondrial membrane [186]. The most important feature of CoQ_10_ is the strong antioxidant capacity of its redox forms (ubiquinone, semi-ubiquinone, ubiquinol) that co-exist in the mitochondrial membrane. CoQ_10_ exhibits anti-lipogenic properties and thus may have a positive effect on NAFLD. It has been proposed that CoQ_10_ may regulate hepatic lipid metabolism by inducing AMPK activation (inhibition of adipogenesis and activation of fatty acid oxidation), thereby inhibiting excessive accumulation of hepatic lipids and preventing NAFLD progression [187]. In addition, CoQ_10_ supplementation was shown to improve liver biochemical enzymes in NASH-induced mice. CoQ_10_ may modulate TNF-α and adiponectin levels to ameliorate inflammation and improve lipid homeostasis in the liver [188]. For example, in young swimmers, intake of CoQ_10_ (300 mg/day for 12 days) reduced oxidative stress and the plasma levels of inflammatory markers such as TNF-α and IL-6, and increased the activity of antioxidant enzymes. Similarly, dietary CoQ_10_ (during 26 weeks of diet supplemented with 0.07–0.7% (*w*/*w*) CoQ_10_) induced the reduction of plasma OS and inflammatory markers in a murine model of the metabolic syndrome [189].

The mechanisms and roles of antioxidants in the treatment of NAFLD have been summarized in Table 1.

### 6.6. Diet

Insulin resistance associated with diets, especially high-fat or high-carbohydrate diets, increase oxidative stress and the risk of NAFLD [202]. Saturated fatty-acids-enriched foods, not unsaturated fatty acids, such as processed meat or high-fat dairy, induce fat accumulation and insulin resistance, which promotes NAFLD progression [203]. NAFLD patients prefer to follow Western eating habits. Obviously, improving the diet style may be the fundamental way to prevent NAFLD to NASH and hepatic cirrhosis through the metabolic homeostasis connected by the liver–gut axis.

The intake of unsaturated fatty acids instead of saturated fatty acids and isocaloric foods with lower fat content can reduce effectively fat deposition in the liver. In addition, the European Society for Clinical Nutrition and Metabolism (ESPEN) and American Gastroenterology Association (AGA) propose the Mediterranean-style diet [204,205]. The Mediterranean diet is evaluated as a diet style that is beneficial to NAFLD. It did not increase intrahepatic lipid content and transaminase levels, although it is iso-caloric or there are no changes in body weight [206,207]. Remarkably, The Mediterranean diet is characterized by rich antioxidants and fiber-balanced lipid and low sugar content [208]. As we have shown above, the antioxidants in the Mediterranean diet effectively protect the liver from oxidative stress, which has a negative effect on the development of NAFLD. Apart from healthy diets, reasonable exercise seems indispensable. An individual lifestyle with a healthy diet and increased exercise are general interventions in reversing NAFLD progression [204,205]. Shortly, a reasonable diet, including rich fruits, vegetables, and protein, is effective against fat-mediated oxidative damage.

## 7. Conclusions

NAFLD is characterized by the accumulation of excess fat in the liver, which leads to liver damage. Fat accumulation in NAFLD is not caused by alcohol consumption. Hepatocytes are the main parenchymal cells in the liver, accounting for about 80% of the liver mass. Non-parenchymal cells include resident KCs, HSCs, liver sinusoidal endothelial cells, and other immune cells. These cells contribute to steatosis, inflammation, and fibrosis and promote the progression of NAFLD to NASH and HCC.

Redox reactions are essential for cell survival and for the maintenance of normal liver function. Due to excess fat intake in NAFLD, there is a significant derailment of redox reactions. Energy metabolism occurs primarily in mitochondria; abnormal redox reactions lead to energy metabolism disorders and subsequently increase mitochondrial dysfunction. ROS are increased during mitochondrial oxidative damage and are formed as byproducts of redox reactions. The production of ROS is not limited to mitochondria, and the ER and peroxisome also generate ROS. The upregulation of β-oxidation reactions due to fatty acid overload increases the production of ROS. However, the key mechanism of NAFLD involves OS-induced damage to the hepatocytes, KCs, and HSCs.

Hepatocytes, KCs, and HSCs contribute to the development of NAFLD. ROS promotes lipid accumulation in hepatocytes through a variety of signaling pathways. The defense mechanisms in hepatocytes (including the antioxidant system and mitophagy) are unable to clear ROS and the damaged mitochondria in a timely manner. mtDNA released by hepatocytes subjected to oxidative damage activates pro-inflammatory KCs through the cGAS/STING pathway. Additionally, inflammatory responses in KCs are also induced by intracellular ROS. This is an important mechanism for the progression of NAFLD to NASH. TGF-β released by damaged hepatocytes and activated KCs enhances fibrogenesis in HSCs. ROS are also involved in the activation of HSCs. In general, the OS triggered by ROS is critical for the development of NAFLD. Therefore, antioxidant therapy may be highly effective for the treatment of NAFLD.

In this review, we summarized reports on the application of classic antioxidants such as flavonoids, curcumin, vitamin E, metformin, and coenzyme Q10 in NALFD models. Although they target different signaling pathways, all these antioxidants ameliorate the pathological changes in NAFLD by reducing ROS. However, the most effective treatment for NAFLD may be to improve the quality of food consumed.

Finally, we also reviewed the role of OS in the development of NAFLD and progression to NASH and hepatic fibrosis and the application of antioxidants to NAFLD.

## Figures and Tables

**Figure 1 antioxidants-11-00091-f001:**
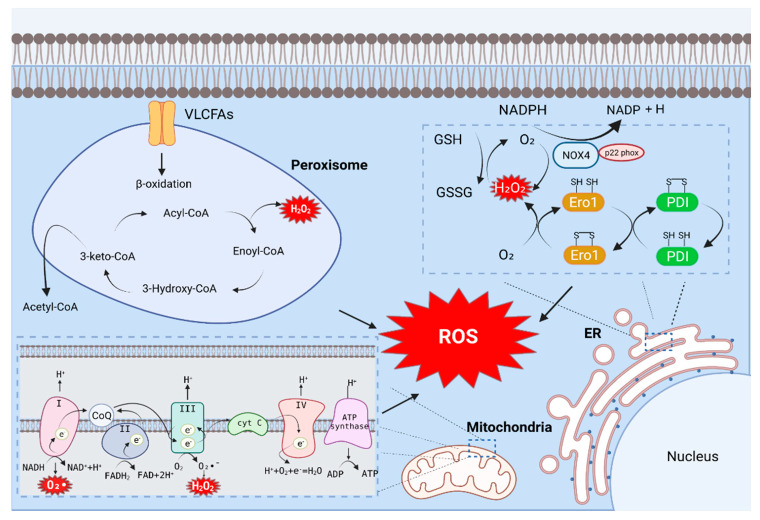
**Major source of ROS production in Hepatic cell.** The catalytic process of oxidoreductase Ero1 and NADPH oxidase (NOX) in the endoplasmic reticulum generates ROS. During cellular respiration, mitochondria transfer electrons to oxygen and generate ROS as a byproduct of oxidative oxidation. Mitochondrial ROS are mostly produced during oxidative phosphorylation of electron transport chains (ETCs) present in the inner mitochondrial membrane. The main cause of ROS generation in peroxisomes is peroxisomal β-oxidation. Acyl-CoA is converted to Enoyl-CoA by acyl-CoA oxidase containing FAD, which provides electrons directly to oxygen to produce H₂O₂.

**Figure 2 antioxidants-11-00091-f002:**
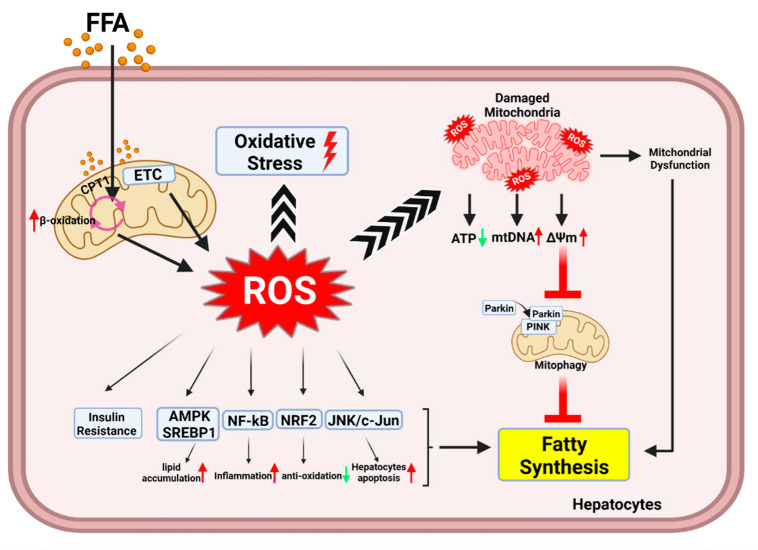
**The mechanism of ROS-mediated oxidative stress and lipid metabolism in hepatocytes.** In the NAFLD patients, the overload intake of free fatty acids increases fatty acid β-oxidation and electron transport chain activity in the mitochondria. This ultimately leads to an increased release of ROS as byproduct of metabolism. The lots of ROS can directly target mitochondria, resulting in decreased energy metabolism, increased release of mtDNA and mitochondrial dysfunction. At the same time, mtROS leads to disturbance of mitochondrial membrane potential. High mitochondrial membrane potential is not friendly to the occurrence of mitophagy. The reduction of mitophagy reduces the clearance of damaged mitochondria and indirectly increases the fat synthesis. Moreover, high levels of ROS activate AMPK, SREBP1, NF-kB, JNK/cJun, and downregulate NRF2. These protein kinases and transcription factors regulate lipid metabolism, inflammation, antioxidant capacity and hepatocytes apoptosis. Finally, with the increase in insulin resistance, the synthesis of fat is intensified in hepatocytes.

**Figure 3 antioxidants-11-00091-f003:**
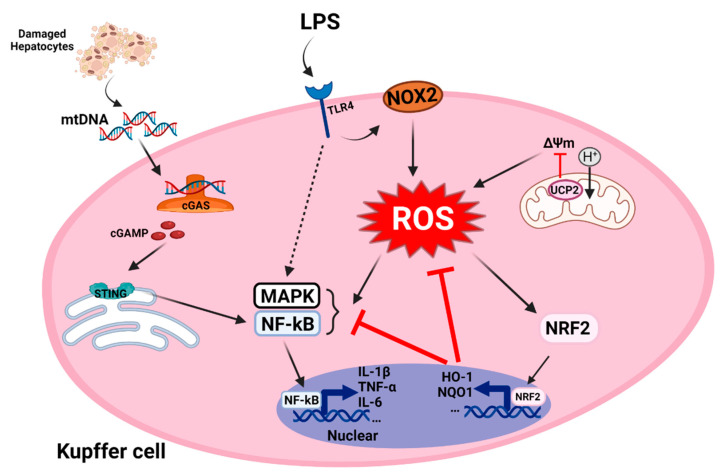
**The mechanism of ROS involved in pro-inflammatory Kupffer cells.** Fatty acid intake increases lipid accumulation in hepatocytes, whereas oxidative stress involves the release of mtDNA from damaged hepatocytes. KCs activated by mtDNA promotes the secretion of pro-inflammatory factors through the cGAS/STING axis and is dependent on the NF-kB pathway. LPS-mediated inflammation through activation of MAPK and NF-kB pathways. Meanwhile, LPS activates TLR4 to promote NOX2 production of ROS. Large amounts of ROS increase MAPK and NF-kB transcription and promote production of inflammatory cytokines as well as (IL-1β, TNF-α, and IL-6, etc.). Mitochondria are the main organelle for ROS production, and UCP2 localized in the inner mitochondrial membrane increase proton transfer to downregulate mitochondrial membrane potential to control ROS levels. NRF2 regulate negatively ROS by increasing the production of antioxidant factors.

**Figure 4 antioxidants-11-00091-f004:**
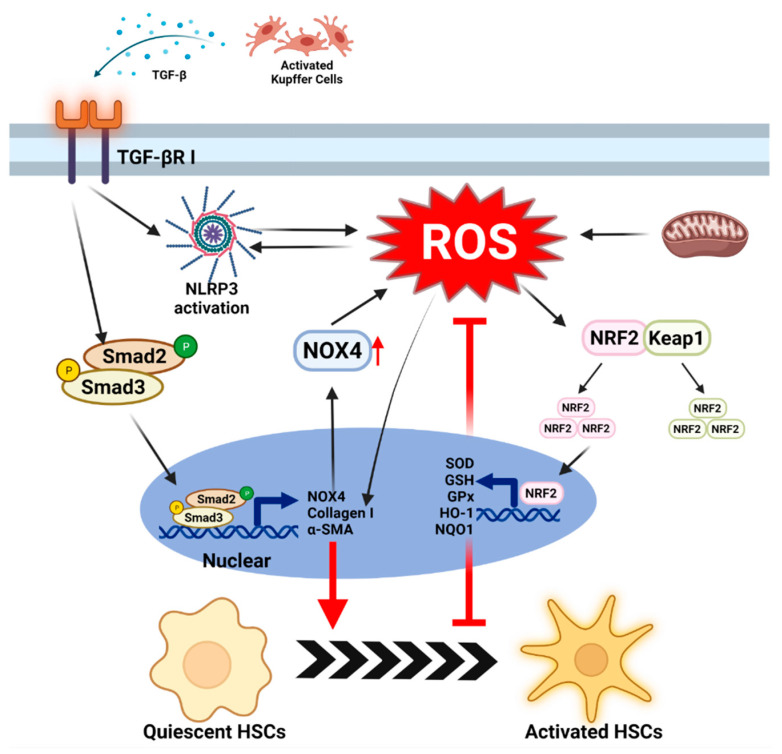
**The mechanism of ROS in activation of stellate cells.** TGF-β is released from activated KCs, the major cell type for cytokine production during NASH. the TGF-β-SMAD2/3 axis is activated in HSCs, which upregulates NOX4 expression to increase ROS production. The additional ROS directly drives the production of extracellular matrix (collagen-I and α-SMA) in HSCs. On the other side, TGF-β elevate NLRP3 activation to leads ROS production. The concomitant release of ROS from mitochondria exacerbates the activation of HSCs. Although moderate amounts of ROS can activate the dissociation of NRF2 and KEAP1 and drive NRF2 transcription, the NRF2-regulated antioxidant system can hardly resist the impact of large amounts of ROS in cells.

**Table 1 antioxidants-11-00091-t001:** Therapeutic compounds based on antioxidant properties in NAFLD.

Name	Mechanism	Effects	CLD	Stage	Ref.	PubChem CID
** *α* ** **-tocopherol** **Atorvastatin**	Inhibition of MAPK (JNK/p38 MAPK) signaling and NF-κB activation.	Free radical stabilization Replenish liver glutathione, reduce steatosis, improve inflammation, hepatic stellate cell activation, collagen mRNA expression and fibrosis.	NASH	Clinical phase 2	[190] [191] [192] [193]	14985
	Inhibition of NF-Κb signaling and HMG-CoA reductase.	Reduction activity of fatty acid β-oxidation, activation of ChREBP, plasma ALT levels, and inflammation in liquid fructose-fed mice.	NAFLD NASH	Clinical phase 2	[194] [195] [196]	60823
**Epigallocatechin-3-gallate (EGCG)**	Inhibition of oxidative stress Inhibition of NF-κB/AKT signailing Inhibition of TGF/ SMAD signaling	Increased SOD activity in serum and liver tissue. Reduction of liver fibrosis in NASH rodent model. Reduction TNF-a and IL-6 mRNA expression in serum and liver tissue.	NASH	Clinical phase 1	[168] [169] [170]	65064
**Metformin**	AMPK and NRF-2 signaling activation	Inhibition of ROS production Increase in SOD2 activation Decrease lipid accumulation in liver and TG level in serum	NAFLD NASH	Clinical phase 4	[182] [183] [184]	4091
**Coenzyme Q**	AMPK signaling activation	Inhibition of NADPH oxidase expression Inhibition of liver steatosis Reduction of TNF-a, IL-6 mRNA expression	NAFLD NASH	Clinical phase	[187] [188] [189] [197]	5281915
**Curcumin**	AMPK/ACC signaling activation	Induction of TAC, GSH-Px and SOD level Inhibition lipid accumulation in liver Inhibition hepatic inflammation and fibrosis Reduction of ER stress	NAFLD NASH	Clinical phase 4	[174] [175] [198]	969516
**Resveratrol**	Activation of autophagy Inhibition of NF-Κb signaling	Induction of superoxide dismutase(SOD), glutathione peroxidase and catalase Regulation of lipid accumulation through autophagy Inhibition of hepatic inflammation	NAFLD, NASH	Clinical phase 2	[199] [200] [201]	445154

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
