# Peer review of "Oxidative Stress Is a Key Modulator in the Development of Nonalcoholic Fatty Liver Disease"

_antioxidants, 2021, doi:10.3390/antiox11010091_

Round 1
Reviewer 1 Report
The work is a review - and there are many related to NFLD. However, this one sums up well what has been done over the past 10 years. However, I think the topic of NFLD and oxidative stress is well chosen as this particular topic has not been covered too often. I just have one specific comment: the authors may want to consider how diet can affect NFLD. More importantly, however, how patients can reduce oxidative stress in the liver by following certain diets. I feel like the authors want to add a little paragraph on this topic.
Minor comments - please verify - there are a number of typos
The following references shall be included and discussed:
- https://doi.org/10.1111/liv.14360
- https://doi.org/10.1179/1351000213Y.0000000050
- JHEP Reports 2021 vol. 3 j 100346
Author Response
Thank you for your insightful comments.
Please see the attachment.

Reviewer 2 Report
Dear Editor,
Overall recommendation:
Accept
Final comments:
In this manuscript, the authors showed a comprehensive mechanisms of oxidase mediated NASH fibrosis and therapeutic applications. Moreover, the paper is well-written and the clinical applications seem appropriate. I think this paper is good for publication in the present form.
Kanasai Medical University
Katsunori Yoshida
Author Response

(The authors gave the same response as above.)

Reviewer 3 Report
The Authors of the paper have reviewed the causes of reactive oxygen species in cells, the relationship between reactive oxygen species (ROS) and nonalcoholic fatty liver disease (NAFLD), and antioxidants as therapeutic agents for NAFLD. They have analyzed issues: factors contributing to oxidative stress in liver, mechanisms of oxidative stress in hepatic steatosis, mechanisms of oxidative stress of Kupffer cells in nonalcoholic steatohepatitis, mechanisms of oxidative stress in hepatic fibrosis and antioxidants as treatments for NAFLD (flavanols, curcumin, vitamin E, metformin, coenzyme Q10)
This manuscript is a significant contribution to the scientific discussion about oxidative stress promotes non-alcoholic fatty liver disease.
Text and table editing and minor language revisions should be made.
I recommend it for publication after minor revision.
Author Response

(The authors gave the same response as above.)
